# Co-designing organisational improvements and interventions to increase inpatient activity in four stroke units in England: a mixed-methods process evaluation using normalisation process theory

David Clarke [ID],[1] Karolina Gombert-Waldron,[2] Stephanie Honey,[3] Geoffrey Cloud,[4] Ruth Harris [ID],[5] Alastair Macdonald,[6] Christopher McKevitt,[7] Glenn Robert [ID],[5] Fiona Jones[8]

► Prepublication history and supplemental material for this paper is available online. To view these files, please visit the journal online (http://dx.doi.org/10.1136/bmjopen-2020-042723).

For numbered affiliations see end of article.

**Correspondence to**
Dr David Clarke;
d.j.clarke@leeds.ac.uk

## ABSTRACT

**Objective** To explore facilitators and barriers to using experience-based co-design (EBCD) and accelerated EBCD (AEBCD) in the development and implementation of interventions to increase activity opportunities for inpatient stroke survivors.

**Design** Mixed-methods process evaluation underpinned by normalisation process theory (NPT).

**Setting** Four post-acute rehabilitation stroke units in England.

**Participants** Stroke survivors, family members, stroke unit staff, hospital managers, support staff and volunteers. Data informing our NPT analysis comprised: ethnographic observations, n=366 hours; semistructured interviews with 76 staff, 53 stroke survivors and 27 family members pre-EBCD/AEBCD implementation or post-EBCD/AEBCD implementation; and observation of 43 co-design meetings involving 23 stroke survivors, 21 family carers and 54 staff.

**Results** Former patients and families valued participation in EBCD/AEBCD perceiving they were equal partners in co-design. Staff engaged with EBCD/AEBCD, reporting it as a valuable improvement approach leading to increased activity opportunities. The structured EBCD/AEBCD approach was influential in enabling coherence and cognitive participation and legitimated staff involvement in the change process. Researcher facilitation of EBCD/AEBCD supported cognitive participation, collective action and reflexive monitoring; these were important in implementing and sustaining co-design activities. Observations and interviews post-EBCD/AEBCD cycles confirmed creation and use of new social spaces and increased activity opportunities in all units. EBCD/AEBCD facilitated engagement with wider hospital resources and local communities, further enhancing activity opportunities. However, outside of structured group activity, many individual staff–patient interactions remained task focused.

**Conclusions** EBCD/AEBCD facilitated the development and implementation of environmental changes and revisions to work routines which supported increased

## Strengths and limitations of this study

► This process evaluation reports the first use of experience-based co-design (EBCD) and accelerated EBCD (AEBCD) in stroke services.

► Analysis informed by normalisation process theory before, during and after use of EBCD/AEBCD provided for an in-depth understanding of staff engagement, local organisational contexts, the impact of co-designed changes on day-to-day working practices of stroke unit staff and the experiences of stroke survivors both as inpatients and as participants in EBCD/AEBCD.

► Recruitment of stroke survivors and family members to participation in EBCD/AEBCD activities was good across all sites but it proved more difficult to recruit former inpatient stroke survivors to participate in post-EBCD/AEBCD evaluation interviews.

► The process evaluation was not designed to generate data to evaluate the longer term sustainability of interventions developed to increase activity opportunities for inpatient stroke survivors.

► Researchers undertaking the process evaluation were part of the core research team for the Collaborative Rehabilitation in Acute Stroke Study and not a separately employed process evaluation team.

activity opportunities in stroke units providing post-acute and rehabilitation care. Former stroke patients and carers contributed to improvements. NPT's generative mechanisms were instrumental in analysis and interpretation of facilitators and barriers at the individual, group and organisational level, and can help inform future implementations of similar approaches.

## INTRODUCTION

Stroke is the second most common cause of death worldwide[1] and is associated with

significant long-term disability.[2] Specialist rehabilitation in stroke units contributes substantially to regaining independence.[3] However, observational studies in stroke units identify high levels of inactivity. In studies spanning 40 years, patients have been reported to be physically active between 13% and 23% of the waking day, and engaged in cognitive or social activity between 4% and 32% of the waking day.[4–10] In these studies, activity related largely to participation in planned therapy, typically physiotherapy (PT), occupational therapy (OT), speech and language therapy (SLT).[11–14] Outside of planned therapy, patients report being bored and can be alone and inactive for 60% of the day.[7 15 16]

This seemingly intractable problem was addressed in two early studies in England. Following observation of activity levels across four rehabilitation wards for the elderly, Ellul et al introduced individualised (physical) activity programmes on two wards facilitated by nurses and scheduled social group activities with staff acting as activity leads; in total 51 patients were observed pre-intervention and post-intervention.[17] Similarly, in a single rehabilitation unit across a period of 2 years, Newall et al introduced regular leisure activity service visits, access to computers, weekly discussion groups, communal lunches and group meetings with other stroke survivors, and encouraged family involvement in therapy practice; activity levels of 67 patients were observed during this time period.[18] Both studies reported increases in useful activity and decreased time spent passively at the bedside. However, neither study led to wider adoption of the approaches reported. While post-stroke inactivity can contribute to poorer outcomes, increased participation in structured physical activity is associated with improved physical function[19] and greater independence.[20] Evidence suggests increasing social and cognitive activity may reduce burden associated with post-stroke mood disorders and cognitive impairment.[21] Interest is increasing in the potential of environmental enrichment (EE)[8 10 22–25] to increase physical, social and cognitive activity in stroke units. EE is defined as 'interventions designed to facilitate physical (motor and sensory), cognitive and social activity by provision of equipment and organisation of a stimulating environment'[23] (p48). There is also interest in the impact of the built environment on activity and social interaction.[26 27] However, across these studies, there remains limited evidence of sustained change in inpatient activity.[22–25]

To date, approaches to address inactivity post-stroke have been largely externally designed and researcher led with limited or no involvement of stroke survivors, caregivers or staff in study design or delivery. As an alternative, participatory improvement approaches involve directly engaging service users and providers in a collaborative process to 'co-produce' a service that addresses the needs and wants of stakeholders while ensuring the improved service can be delivered using existing resources.[28–32] Co-production approaches provide a means for stakeholder voices not only to be heard in terms of improving services but also provide a framework for stakeholder participation throughout the improvement process.[29 32–34] Specific forms of co-production such as experience-based co-design (EBCD)[29 35–37] have been developed and applied in healthcare. The Collaborative Rehabilitation in Acute Stroke (CREATE) Study evaluated the impact of using the six-stage EBCD approach in two stroke units and an accelerated version (AEBCD)[38 39] in two further units (figure 1) to increase patients' social, cognitive and physical activity. The CREATE Study findings are reported elsewhere[38 39]; in summary, qualitative findings indicated it was feasible to co-produce changes in all four stroke units to increase opportunities for social, cognitive and physical activity through joint work in three priority areas: 'space' (environment), 'activity' and 'communication'. Patients, families and staff perceived positive benefits from participating in EBCD/AEBCD. However, quantitative data did not demonstrate consistent increases in physical, social or cognitive activity.

Despite extensive use of EBCD internationally in the last 10 years, there has been limited evaluation of the process and outcomes of the approach, and none in stroke.[36 40] The use of EBCD/AEBCD in the CREATE Study to develop and implement quality improvements is typical of a complex intervention involving 'multiple components which interact to produce change'.[41] Medical Research Council[41 42] guidance for evaluation of complex interventions recommends use of process evaluations to explore not only 'whether interventions 'worked' but (also) how they were implemented, their causal mechanisms and how effects differed from one context to another'.[42] This paper reports on the embedded process evaluation in the CREATE Study. The aim of the process evaluation was to explore facilitators and barriers to using EBCD and AEBCD in the development and implementation of improvements to increase activity opportunities for inpatient stroke survivors.

## METHODS
### Patient and public involvement
Listening to and using patients' and carers' voices, experiences and ideas is central to EBCD[29] but patients and the public were also involved from study inception, participating in development of the research proposal which was discussed with stroke survivors and carers at two stroke research group meetings. There was strong support for the research, particularly the participatory approach planned; these groups provided important insights about how to facilitate participants' involvement including being aware of post-stroke fatigue, challenges of access to hospitals sites, running EBCD/AEBCD events on or near to stroke units and ensuring transport, parking, access and expenses for involvement were considered in advance of onsite participation. A younger stroke survivor and carer became members of the Study Steering Committee (SSC), participating in review of participant information, discussions about conducting observations, interviews and co-design meetings with patients and staff.

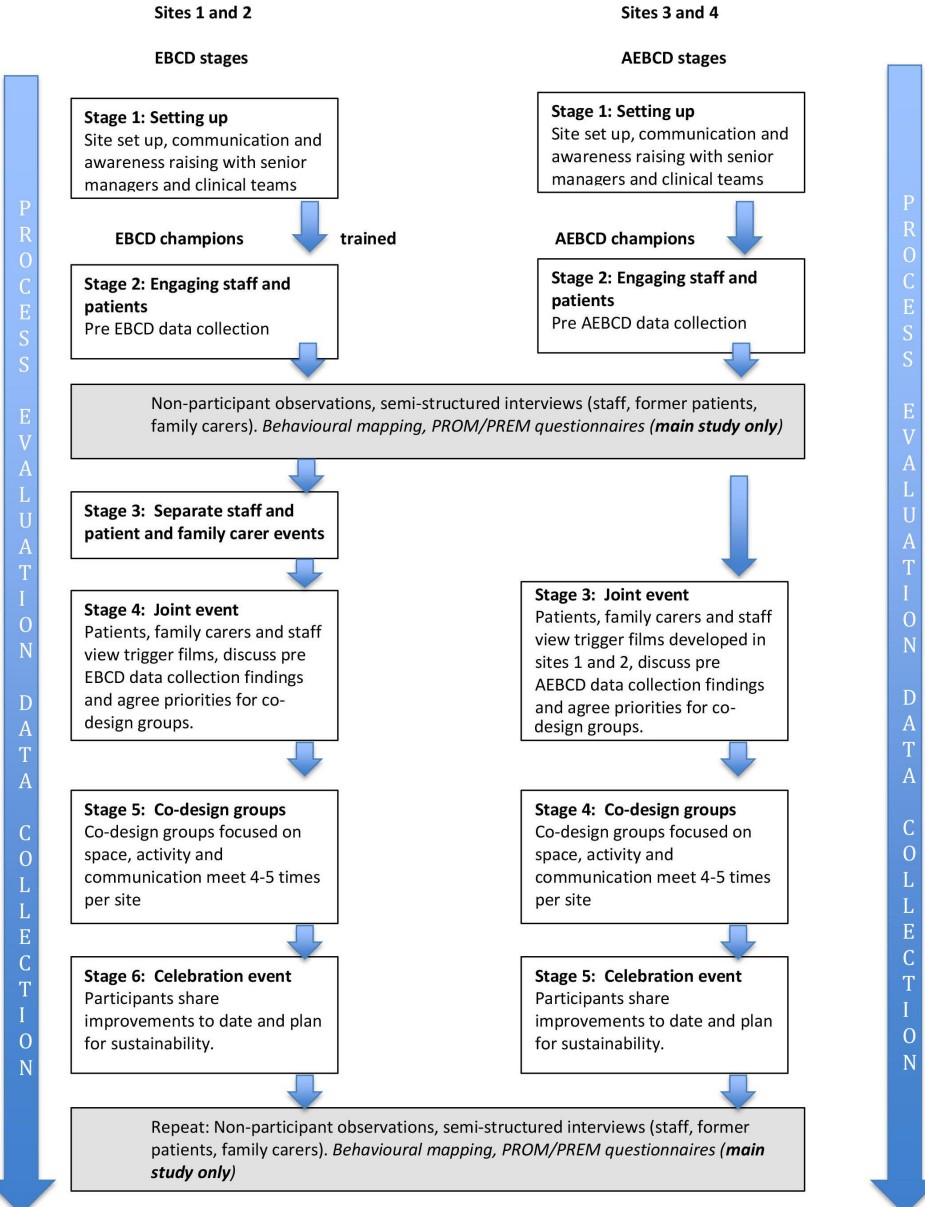

Sites 1 and 2

**EBCD stages**

**Stage 1: Setting up**
Site set up, communication and awareness raising with senior managers and clinical teams

EBCD champions          trained

**Stage 2: Engaging staff and patients**
Pre EBCD data collection

Non-participant observations, semi-structured interviews (staff, former patients, family carers). *Behavioural mapping, PROM/PREM questionnaires (**main study only**)*

**Stage 3: Separate staff and patient and family carer events**

**Stage 4: Joint event**
Patients, family carers and staff view trigger films, discuss pre EBCD data collection findings and agree priorities for co-design groups.

**Stage 5: Co-design groups**
Co-design groups focused on space, activity and communication meet 4-5 times per site

**Stage 6: Celebration event**
Participants share improvements to date and plan for sustainability.

Sites 3 and 4

**AEBCD stages**

**Stage 1: Setting up**
Site set up, communication and awareness raising with senior managers and clinical teams

AEBCD champions

**Stage 2: Engaging staff and patients**
Pre AEBCD data collection

**Stage 3: Joint event**
Patients, family carers and staff view trigger films developed in sites 1 and 2, discuss pre AEBCD data collection findings and agree priorities for co-design groups.

**Stage 4: Co-design groups**
Co-design groups focused on space, activity and communication meet 4-5 times per site

**Stage 5: Celebration event**
Participants share improvements to date and plan for sustainability.

PROCESS EVALUATION DATA COLLECTION

Repeat: Non-participant observations, semi-structured interviews (staff, former patients, family carers). *Behavioural mapping, PROM/PREM questionnaires (**main study only**)*

**Figure 1** CREATE Study design and methods with embedded process evaluation. AEBCD, accelerated EBCD; CREATE, Collaborative Rehabilitation in Acute Stroke; EBCD, experience-based co-design; PREM, patient-reported experience measure; PROM, patient-reported outcome measure.

They helped shape the messaging in EBCD/AEBCD feedback events, reviewed emerging findings and commented on researchers' summary explanations of findings.

## CREATE Study design and methods

CREATE used a mixed-methods case-comparison design with embedded process evaluation[38 39] (figure 1). EBCD[29 31 35] or AEBCD[37] was introduced into four stroke units (sites 1–4), two in London and two in the North of England (Yorkshire).

Written site consent for data collection was provided by the principal investigator at each site. For non-participant observations, we used a verbal process consent approach, checking that staff, family carers and stroke survivors agreed to observations; no individual data were recorded. We gained written informed consent from all stroke

survivors, family carers and staff participating in interviews and EBCD/AEBCD activities.

## Contexts

Stroke units were purposively selected, focusing on units not already taking part in large clinical trials, that could commit to participation in a multistage study over 6–9 months. Sites were stroke units receiving patients after care in hyperacute units in the same hospital (sites 2 and 4) or major stroke centres (sites 1 and 3) (table 1). Biannual National Acute Stroke Organisational Audit data[43] indicated the units performed within the mid-range across key quality indicators and were subject to the same staffing pressures and increasing caseload complexity reported nationally. Clinical leads at each site acted as principal investigators or supported the study.

**Table 1**  Unit characteristics

|  | Site 1 (EBCD) | Site 2 (EBCD) | Site 3 (AEBCD) | Site 4 (AEBCD) |
|---|---|---|---|---|
| Number of stroke beds | 24 | 24 | 26 | 26 beds across two adjoining wards (14 and 12 beds) |
| Hospital type | District general hospital with 629 beds | District general hospital with 500 beds | District general hospital with 700 beds | District general hospital with 600 beds |
| Number of stroke patients treated per year | 195 | 978* | 250 | 640* |
| Typical length of stay (days) | 28 | 13* | 28 | 21* |
| 7-day therapy service | No | An OT and PT worked on Saturday covering the acute and rehabilitation units; a stroke rehabilitation assistant worked on the rehabilitation unit on Saturdays and Sundays | No | No |
| Performance in National Acute Organisational Audit (RCP, 2017) | Achieved 7 of the 10 key indicators | Achieved 4 of 10 key indicators | Achieved 8 of the 10 key indicators | Achieved 5 of the 10 key indicators |

*Data for sites 2 and 4 include data for hyperacute/acute units in the same hospital.
AEBCD, accelerated EBCD; EBCD, experience-based co-design; OT, occupational therapist; PT, physiotherapist; RCP, Royal College of Physicians.

## Participants

As part of the embedded process evaluation, semistructured interviews were conducted with 76 staff, 53 stroke survivors (previously inpatients) and 27 family carers pre-EBCD/AEBCD implementation or post-EBCD/AEBCD implementation. Forty-three co-design meetings were held across sites involving 23 stroke survivors, 21 family carers and 54 staff including PTs, OTs, SLTs, dietitians, nurses, rehabilitation and healthcare support workers (HCSWs), hospital managers, support staff and volunteers. A total of 366 hours of ethnographic observations were completed (table 2).

## Process evaluation

Normalisation process theory (NPT)[44 45] underpinned the process evaluation which encompassed sites preparation for and use of EBCD/AEBCD. NPT is concerned with understanding how complex interventions are implemented and integrated into existing healthcare systems and is conceptualised through four generative mechanisms each with four components (table 3). These mechanisms represent what participants 'do' to get the work of implementation done. They can be understood as a process (not necessarily linear) in which participants make sense of a new or different way of working, commit to it, make the effort required to work in that way and undertake continuous evaluation. NPT was used in two ways, first to guide data generation at each site and second as a sensitising lens in ongoing data analysis. NPT's constructs were used to identify and reflect on processes that may act as facilitators or barriers to using EBCD/AEBCD to

**Table 2**  Number of participants by data generation method

| Site | Staff interviews | Patient interviews | Carer interviews | Non-participant observations (hours) |
|---|---|---|---|---|
| Site 1 pre-EBCD | N=13 | N=9 | N=4 | 50 |
| Site 1 post-EBCD | 8 | 5 | 5 | 47 |
| Site 2 pre-EBCD | 15 | 9 | 4 | 48 |
| Site 2 post-EBCD | 7 | 6 | 2 | 44 |
| Site 3 pre-AEBCD | 6 | 9 | 3 | 50 |
| Site 3 post-AEBCD | 8 | 6 | 3 | 37 |
| Site 4 pre-AEBCD | 7 | 4 | 3 | 44 |
| Site 4 post-AEBCD | 12 | 5 | 3 | 46 |
| Total | 76 | 53 | 27 | 366 hours |

AEBCD, accelerated EBCD; EBCD, experience-based co-design.

**Table 3** Normalisation process theory (NPT)

| NPT constructs | Components | Explanation |
|---|---|---|
| Coherence | ▶ Differentiation<br>▶ Communal specification<br>▶ Individual specification<br>▶ Internalisation | The sense-making work that people do individually and collectively when faced with implementing changes to existing working practices. This would include differentiating new practices from existing work and thinking through not only the perceived value and benefits of desired/planned changes but also what work will be required of individual people in a setting to bring about these changes. |
| Cognitive participation | ▶ Initiation<br>▶ Enrolment<br>▶ Legitimation<br>▶ Activation | The work that people need to do to engage with and commit to a new set of working practices. This often requires bringing together those who believe in and are committed to making changes happen. This also involves people working together to define ways to implement and sustain the new working practices. |
| Collective action | ▶ Interactional workability<br>▶ Relational integration<br>▶ Skill set workability<br>▶ Contextual integration | The work that will be required of people to actually implement changes in practices, including preparation and/or training of staff. Often this entails rethinking how far existing work practices and the division of labour in a setting will have to be changed or adapted to implement the new practices. This requires consideration of not only who will do the work required, but also the skills and knowledge of people who will do the work and the availability of the resources they need to enact and sustain the new working practices. |
| Reflexive monitoring | ▶ Systematisation<br>▶ Communal appraisal<br>▶ Individual appraisal<br>▶ Reconfiguration | People's individual and collective ongoing informal and formal appraisal of the usefulness or effectiveness of changes in working practices. This involves considering how the new practices affect the other work required of individuals and groups, and whether the intended benefits of the new working practices are evident for the intended recipients and staff. |

develop and implement improvements to increase activity opportunities for inpatient stroke survivors.

### Process evaluation data sources

A data generation plan linked to NPT's constructs identified data to be generated pre-EBCD/AEBCD and post-EBCD/AEBCD processes in sites (online supplemental file 1). In each site, data were generated by researchers pre-implementation and post-implementation of EBCD/AEBCD cycles through ethnographic non-participant observations of patient and staff activity at different times of the day and at weekends. Observations were conducted over a period of 10 days, in sites 1 and 3 by FJ, KG-W and AC, in sites 2 and 4 by DC and SH. Observation periods typically lasted 4–5 hours; each researcher completed 3–4 observation periods in each unit pre-implementation and post-implementation. Semistructured audio or video interviews were conducted by FJ, TK, AC and KG-W in sites 1 and 3, and by SH and DC in sites 2 and 4 with a purposive sample of stroke unit staff, stroke survivors who had been inpatients in the units in the 6 months prior to EBCD/AEBCD cycles and family carers. Interviews were repeated post-EBCD/AEBCD implementation. Researchers' observations of staff training in EBCD/AEBCD, reflections on facilitating EBCD/AEBCD cycles, and on informal and formal engagement with participants in sites as part of recruitment activity and through observations and interviews were included. See online supplemental file 2 for demographic data.

### Process evaluation data analysis
#### Data analysis

Qualitative data were managed in NVivo V.10.[46] Using a process of ongoing integrative analysis,[47] themes were identified and reviewed at each site and then discussed by FJ, DC, KG-W and SH in three monthly face-to-face meetings, followed by review of the full data set. Analysis was underpinned by use of NPT's constructs and subcomponents. Once EBCD/AEBCD activities ceased in all sites, summary memos with researcher reflections were used to construct a single integrated account. Confirmability occurred through independent, then joint and team half-day analysis cycles, followed by discussing emerging findings with SSC members. Credibility and transferability are evident in the use of data extracts to support explanations of observational and interview data in terms of participants' engagement with EBCD/AEBCD, as well as the facilitators and barriers to developing and implementing increased activity opportunities in sites.

### Process evaluation results

Data generated through non-participant observations, semistructured interviews and evaluations from EBCD/AEBCD events indicated positive experiences of participation in EBCD/AEBCD. Most participants perceived EBCD/AEBCD processes led to changes which were increasing opportunities for independent and supervised activity in all sites. These changes affected not only inpatient stroke survivors' experiences but also those of their family carers and stroke unit staff. NPT mechanisms rarely operate in a linear or standalone way; in table 4 and below we highlight the combinations of mechanisms we identified occurring within and across sites during the EBCD/AEBCD process and which in turn were identified as directly and indirectly helping to increase activity opportunities.

**Table 4** NPT mechanisms and components identified during the EBCD/AEBCD process

| Observed processes | Combinations of NPT mechanisms and components identified during the EBCD/AEBCD process | Facilitators influencing use of EBCD/AEBCD as a process to develop and implement improvements to increase activity opportunity | Barriers and challenges to using EBCD/AEBCD as a process to develop and implement improvements to increase activity opportunity |
|---|---|---|---|
| Staff, inpatients, former patients and carers made sense of and engaged with EBCD/AEBCD at different rates over time. | Coherence:<br>▲ Differentiation<br>▲ Individual specification<br>Cognitive participation:<br>▲ Initiation<br>▲ Enrolment | ▲ Small groups of staff with access to EBCD/AEBCD team training<br>▲ Researcher presence during observation and interview periods provided; opportunities to explain EBCD/AEBCD to more staff one to one and in small groups<br>▲ Patients, former patients and carers became aware of the study through observations and interview participation. | ▲ Without access to EBCD/AEBCD training –challenging in all sites to facilitate individual and collective understanding and engagement in wider stroke unit and hospital staff and volunteers<br>▲ Delay between training and commencing EBCD/AEBCD activity in all sites- early interest in the study dissipated- impacted on understanding what EBCD/AEBCD process could achieve and would mean for wider staff groups<br>▲ Inpatients did not participate in co-design meetings and had no direct in engagement creating increased activity opportunities. |
| The facilitated EBCD/AEBCD process was perceived to be different and more effective than prior change initiatives designed to improve activity opportunities. | Coherence:<br>▲ Differentiation<br>▲ Individual specification<br>▲ Communal specification<br>Cognitive participation:<br>▲ Initiation<br>▲ Enrolment<br>▲ Legitimation<br>Collective action:<br>▲ Interactional workability | ▲ Structured, time limited and externally facilitated approach of EBCD/AEBCD<br>▲ Sites 1 and 2: large numbers of staff participate in separate and joint meetings Sites 3 and 4 (AEBCD) joint meetings important in engaging staff in the process<br>▲ EBCD/ABCD as a tried and tested service improvement model- legitimised staff time and resource allocation. | ▲ Across all sites limited numbers of stroke unit staff, wider hospital staff, volunteers, former patients and carers can participate in EBCD/AEBCD process activities. |
| Former patient & family member participation in EBCD/AEBCD | Coherence:<br>▲ Individual specification<br>▲ Communal specification<br>Cognitive participation:<br>▲ Initiation<br>▲ Enrolment<br>▲ Legitimation<br>Collective action:<br>▲ Interactional workability | ▲ Trigger films demonstrate commonality and difference in experiences and facilitate former patients and carers identification as a group with shared beliefs on changing patients' activity experiences in their local stroke units.<br>▲ Trigger films are a powerful means of highlighting stroke survivors' and carers' experiences related to inactivity and activity in stroke units-these heighten staff understanding of stroke survivors' individual and collective experiences<br>▲ EBCD/AEBCD engages participants as equal partners in the improvement process. | ▲ Recruitment and retention of a representative range of former patients & family members to participate in EBCD/AEBCD was challenging in some sites; numbers participating were higher at both London sites. |

Continued

**Table 4** Continued

| Observed processes | Combinations of NPT mechanisms and components identified during the EBCD/AEBCD process | Facilitators influencing use of EBCD/AEBCD as a process to develop and implement improvements to increase activity opportunity | Barriers and challenges to using EBCD/AEBCD as a process to develop and implement improvements to increase activity opportunity |
|---|---|---|---|
| Small numbers of staff, volunteers, former patients, family members participate in co-design groups. | Coherence:<br>▲ Communal specification<br>▲ Internalisation<br>Cognitive participation:<br>▲ Enrolment<br>▲ Legitimation<br>▲ Activation<br>Collective action:<br>▲ Interactional workability<br>▲ Relational integration | ▲ Co-design groups were effective with small numbers of regular participants<br>▲ Co-design groups tended to foster participants' continued commitment to individual and collective action and practice change<br>▲ Co-design groups provided a mechanism to involve non stroke unit staff who could suggest and sanction improvements related to estates and community involvement<br>▲ Co-design groups were the drivers of change and improvement | ▲ Wider staff groups had less involvement in planning and actioning change planned by co-design groups<br>▲ Staff numbers varied with many more nurses and HCSWs working in stroke teams than therapists, stroke SRAs and medical staff<br>▲ Strategies for communicating planned change had varied success (newsletters, using routine staff information exchange events, open afternoons)<br>▲ Staff participants in co-design groups had increased workloads, typically with no additional time allocation. |
| Researcher facilitation of EBCD/AEBCD process | Cognitive participation:<br>▲ Initiation<br>▲ Enrolment<br>▲ Legitimation<br>▲ Activation<br>Collective action:<br>▲ Interactional workability<br>▲ Relational integration<br>▲ Skill set workability | ▲ Researcher facilitation of the structured EBCD/AEBCD enabled key staff to commit to participation in co-design groups and to developing and implementing improvements whilst maintaining their usual work roles and responsibilities<br>▲ Researcher facilitation was a key part of recruitment, retention and participation of former patients and carers<br>▲ Researcher facilitation was important in enabling and sustaining involvement of wider hospital staff, volunteers and community involvement<br>▲ Researcher facilitation supported key stroke unit staff in assuming responsibility for improvements after the CREATE study ended. | ▲ Researcher facilitation could mean some staff relied on researchers to lead and progress the EBCD/AEBCD process rather than assuming leadership roles<br>▲ Time required to contact, engage with, involve and sustain participation of non stroke unit staff, volunteers and community groups was initially very high<br>▲ Evidence in other EBCD projects of improvement activity stalling or ceasing when facilitator role ends. |

Continued

**Table 4** Continued

| Observed processes | Combinations of NPT mechanisms and components identified during the EBCD/AEBCD process | Facilitators influencing use of EBCD/AEBCD as a process to develop and implement improvements to increase activity opportunity | Barriers and challenges to using EBCD/AEBCD as a process to develop and implement improvements to increase activity opportunity |
|---|---|---|---|
| The influence of managerial authority and support for stroke unit staff participation and improvement activity | Cognitive participation:<br>▲ Initiation<br>▲ Enrolment<br>▲ Legitimation<br>Collective action:<br>▲ Interactional workability<br>▲ Relational integration<br>▲ Skill set workability<br>▲ Contextual integration | ▲ Managerial support for involvement of staff in the EBCD/AEBCD project was critical to the progression of the process in all sites- both in terms of joint meeting attendance for larger groups of staff and those staff routinely involved in co-design groups<br>▲ Managerial support at clinical directorate and stroke unit level was required for co-design groups to navigate complex NHS Trust organisational structures and, in leveraging resources to enable co-design group ideas to progress<br>▲ Involving managers with experience of service improvement elsewhere in the hospital in co-design groups ensured myths and misconceptions about what changes were possible in the organisation were quickly addressed and corrected. | ▲ Securing direct involvement of non stroke unit managers in a project which did not directly address clinical outcomes or National Clinical Audit targets was challenging in all sites<br>▲ Stroke unit staff were reluctant to take direct action involving estates to effect environmental support or use of volunteers on units in the absence of managerial support at clinical directorate or stroke service level. |
| Challenges related to the duration of the change process and time taken for improvements to be introduced. | Cognitive participation:<br>▲ Legitimation<br>▲ Activation<br>Collective action:<br>▲ Relational integration<br>▲ Skill set workability<br>▲ Contextual integration<br>Reflexive monitoring:<br>▲ Systematisation<br>▲ Communal appraisal<br>▲ Individual appraisal | ▲ Full EBCD cycles took approximately 9 months to complete<br>▲ AEBCD cycles took approximately 6 months to complete | ▲ Some seemingly simple improvements took many weeks to plan, develop and implement for example, improved web access; making iPads accessible and secure; providing new and welcoming signage; facilitating on unit volunteer and community group activities<br>▲ Improvements involving changes in space use, environment or volunteer involvement required negotiation with and sometimes additional funding from managers; this could take long periods of time or be blocked or ignored by managers. |

Continued

**Table 4** Continued

| Observed processes | Combinations of NPT mechanisms and components identified during the EBCD/AEBCD process | Facilitators influencing use of EBCD/AEBCD as a process to develop and implement improvements to increase activity opportunity | Barriers and challenges to using EBCD/AEBCD as a process to develop and implement improvements to increase activity opportunity |
|---|---|---|---|
| Implementing and embedding co-designed changes into stroke unit practice | Collective action:<br>▲ Relational integration<br>▲ Skill set workability<br>▲ Contextual integration<br>Reflexive monitoring:<br>▲ Systematisation<br>▲ Communal appraisal<br>▲ Individual appraisal | ▲ 'Visible' improvements including new spaces for activity and environmental changes helped staff not involved in co-design groups understand the kinds of improvements occurring and led to inclusion of these spaces or environmental changes in the practice of therapists and nurses<br>▲ Introduction and use of co-designed improvements such as the 'Something about me board' and personalising bed spaces with photos and personal objects were reported to prompt more personalised supervised and independent activity opportunities<br>▲ Where new activity opportunities supported or could be added to existing rehabilitation practice these quickly became incorporated in the routines of therapists and nurses<br>▲ Agreed, routine volunteer led activity was recognised as valuable in increasing social and cognitive activity opportunities for stroke survivors-this reduced demand on stroke unit staff and helped ensure volunteers had a defined role and contribution on the unit. | ▲ Involving large and diverse staff groups as in stroke units in developing and implementing improvements proved challenging<br>▲ Some staff felt they were not consulted about improvements which would require their on-going involvement for example, completion of required information on the 'something about me boards'; preparing inpatients for and supporting them in breakfast club activity<br>▲ Ensuring unit wide change, across disciplines, in encouraging and supporting activity opportunities was more difficult than securing uni-disciplinary commitment to increasing activity opportunities<br>▲ Routine volunteer activity on units was variably received-staff need to be confident volunteers have received appropriate stroke specific training for example if involved in communication support or mealtime support for stroke survivors. |

AEBCD, accelerated EBCD; CREATE, Collaborative Rehabilitation in Acute Stroke; EBCD, experience-based co-design; HCSWs, healthcare support workers; NHS, National Health Service; NPT, normalisation process theory; SRAs, stroke rehabilitation assitants.

### Coherence and cognitive participation: making sense of and engaging with EBCD/AEBCD over time

Prior to research activity, members of each stroke team attended EBCD/AEBCD training led by the Point of Care Foundation.[28] These staff, supported by a researcher, were to lead EBCD/AEBCD implementation at their site. Full-day training for sites 1 and 2 introduced the CREATE Study and the theory and practice of EBCD/AEBCD; sites 3 and 4 received half-day training focused on implementation. Participants in AEBCD sites heard firsthand of how sites 1 and 2 were working to increase activity and discussed changes made by these teams.

> it was helpful meeting other people involved in the study on their wards, what they did and how it involved patients. (AEBCD training participant feedback, site 3)

Training afforded these staff the opportunity to start making sense of EBCD/AEBCD and to think individually and collectively about sharing their knowledge and understanding with colleagues, about what activity opportunities were possible and what these would entail practically for the day-to-day work of their colleagues and themselves. Training enabled reflection on work with patients and carers using EBCD/AEBCD, and how it differed from written feedback about patients' and carers' experiences. However, ensuring other stroke unit staff understood and engaged with the EBCD/AEBCD process was more challenging. There was no requirement for cascade training or other forms of knowledge sharing. In addition, changes to the UK Health Research Authority and Research Ethics Committee approval processes led to a delay of 4 months between training site 1 and 2 staff and commencing EBCD activity, early interest quickly dissipated; understanding of what the EBCD approach would mean for wider staff groups' day-to-day work was limited.

However, researcher presence during pre-EBCD/AEBCD observation and interview periods enabled wider groups of staff to engage with and make sense of the project. Staff were curious about data generation methods used including behavioural mapping and observations, researchers used these opportunities to explain EBCD/AEBCD. Patients and carers also became aware of the study through the conduct of observations and behavioural mapping; they sought study information and expressed views on activity outside of therapy.

### Coherence, cognitive participation and collective action: the facilitated EBCD/AEBCD process

In sites 1 and 2, EBCD's structured and facilitated approach was a major factor contributing to staff progressing to a more engaged position, readier to commit to considering how change could happen in their site and to thinking through who needed to be involved. Separate and then joint meetings enabled large numbers of staff to participate in EBCD events (figure 1). They viewed trigger films, heard firsthand accounts of patients' and carers' experiences, worked in small groups with colleagues, former

patients and carers to identify 'touch points', and agreed priorities for action. These were high-energy meetings with shared enthusiasm for change and proved a powerful catalyst for larger groups of staff to share the view that change to stroke unit environments, access to resources, and routine-working practices to increase patient activity was possible. These events led to understanding of the role of co-design groups as being to work on ideas for activity opportunities generated by participants in joint meetings. In sites 3 and 4, starting the AEBCD process at the later joint meeting stage meant knowledge and understanding was more limited initially. However, joint meetings still proved important in engaging staff, particularly as information on how sites 1 and 2 had addressed issues identified in trigger films were shared at these meetings.

> It felt quite exciting—it will be interesting to see how it develops, keen to be involved and contribute. (Staff feedback after joint event, site 3)

Other features of EBCD/AEBCD that facilitated cognitive participation and collective action, mainly in co-design group members, were the defined and time-limited nature of the EBCD/AEBCD process. Clinical leads and service managers understood EBCD/AEBCD was a tried-and-tested service improvement model, this legitimised staff time and resource allocation committed to EBCD/AEBCD and associated improvements. Staff in all sites noted EBCD/AEBCD contrasted with previous attempts to introduce change, which were often 'poorly defined' in terms of timescales, roles and responsibilities, and operated without additional resources. EBCD/AEBCD's participatory approach appeared to add a sense of responsibility for staff to deliver on agreed actions, and not to 'let down' patients and carers they worked with in co-design groups. As commitment to increasing activity opportunities grew, small groups of staff, not all of whom were stroke unit based, worked together to progress actions agreed by co-design groups.

### Cognitive participation and collective action: patient and family member participation in EBCD/AEBCD

EBCD/AEBCD ensures patients and carers express their priorities for change and engages them as equal partners in designing and implementing solutions. In sites 1 and 2, separate patient and carer meetings provided opportunities to explore experiences of stroke and activity/inactivity after stroke. Trigger films demonstrated commonality and difference in experiences and began the process of identification by these former patients and carers as a group, with a shared belief in changing patients' activity experiences in their local stroke units. Although former patients and carers in sites 3 and 4 did not have local trigger film participation in common, they recognised long periods of inactivity described in the site 1 and 2 trigger films and lack of activity opportunities as similar to their own; trigger films validated important issues raised. EBCD/AEBCD provided former patients and carers with the means and confidence to give voice to

their experiences and appeared to help these participants understand how they could work with staff as partners in bringing about improvements.

> We felt able to say what we wanted to say and what we wanted to say has turned out to have a valuable effect, so, yeah, very happy to say that we didn't feel intimidated in any way. (Carer, site 4 post)

Participants in EBCD/AEBCD shared a common experience with their peers and with the staff with whom they exchanged ideas and problem solved in co-design groups. This also appeared true for managers, professional leads and volunteers who were 'external' to the day-to-day work of stroke units but involved in EBCD/AEBCD events. Patients' voices and the authentic examples depicted in trigger films were critical in all staff reorientating their focus to patients' experiences. These events contributed to participants' progression from shared understanding of the importance of change to working together to define ways to implement ideas into existing working practices. This included thinking through how changes to increase activity would be actioned, who would take responsibility and who would routinely deliver these actions. These mechanisms represent progression through cognitive participation and collective action over time. Clinical staff facilitated all co-design groups, patients or carers did not seek to lead; but their contributions were actively pursued, and observational and interview data findings confirm these were influential in improvements that occurred. Examples included designing and painting murals (sites 1 and 3) and redesign of patient and carer facing documentation (sites 2 and 4), actions designed to facilitate increased activity through providing stimulating spaces and information about the purpose of new spaces and activity opportunities.

### Cognitive participation and collective action: wider staff groups have less involvement in planning and actioning change

Co-design groups brought together those who believed in and were committed to making change happen. These small, mainly self-selected groups provided a stimulus and workspace for staff, former patients, family members/carers and volunteers to engage with and commit to planning and actioning changes to increase activity opportunities and drive new working practices. However, in the early months of the study, it was difficult for staff not involved in co-design groups to envisage whether or how increasing patient activity may affect their roles and working practice. Coherence and cognitive participation, linked to commitment to change and comprehension of the possible benefits of changes in individual and collective practice, developed at different rates across wider staff groups at all sites.

Early 'wins' such as a new 'social corner' at site 1 raised awareness among all staff, inpatients and visiting family/carers that change was happening. In terms of collective action, some staff incorporated the new space in therapeutic activity or prompted its use independently or with

family. Similarly, changing a room at site 4 from a wheelchair storage area to a day room was visible evidence of change and led to its routine use for independent and supervised activity. Such 'public' examples enabled wider understanding of how EBCD/AEBCD led to implementation of environmental and practice changes; this stimulated additional discussion among some staff groups about ways to incorporate increasing activity into day-to-day practice:

> There's been staff involved from physiotherapy and occupational therapy, we meet in a morning and we've handed over what progressions have been from the CREATE study. So say for example we're telling them about the updating of the garden and if we need to take anybody to the garden. (Co-design group member, site 4 post)

Communication and engagement across nursing teams working early, late, on weekends, and night duty was more difficult to achieve; observations indicated project information and changes made were not routinely included in nursing handovers at any site; this initially limited coherence and cognitive participation among nursing staff.

### Collective action: challenges in implementing co-designed improvements

One challenge of EBCD/AEBCD is the relatively long time period over which change is planned, implemented and evaluated. Managers largely encouraged staff to participate in EBCD/AEBCD but did not allocate time for participation; staff were encouraged to 'work flexibly'. For nursing staff and HCSWs, attending co-design meetings during shifts was problematic. In site 1 staff sickness and workload challenges meant that three of four EBCD-trained staff did not participate in co-design work. At site 4 two AEBCD-trained staff, both nurses, attended the joint meeting but did not participate directly in co-design groups thereafter. Staff-led co-design groups in all sites experienced similar challenges but other staff volunteered to take responsibility. Enthusiasm for the project across all sites meant some staff attended joint meetings on their days off; and in sites 1–3, some staff participated in co-design meetings on days off or before shifts. Most staff in co-design groups reported completing activities in their own time. Where implementation did not require substantial change in roles and was perceived as enhancing patients' experience consistent with rehabilitation, staff indicated the extra effort was worthwhile.

> I think it's given me a massive workload, I think it's doubled it…to be fair. But, I was committed, I mean I took it on, but I've enjoyed that, I'm glad for the changes. [……] I knew we needed changes, so I was happy to help bring the changes. (Staff, site 4 post)

As changes became visible to wider staff groups and involved patients more regularly, staff appeared more receptive to involving external partners. Examples include complementary therapy from a local health network (site

1), fortnightly singing with local university students (site 2) and musicians from a community arts group (site 3). Changes elicited positive and negative responses. At site 2 some staff complained the 'Something About Me' (patient information) board mounted behind patients' beds appeared without consultation. However, these staff also said they liked what the board was designed to do. Such comments highlight how difficult it is to secure cognitive participation and commitment to collective action in staff who have limited engagement with implementation of co-designed interventions. Despite some of the challenges outlined, as activity opportunities became more evident shifts in staff perceptions and behaviours suggested collective action and reflexive monitoring were becoming established in most units. Overall team members' perceived changes and acknowledged the positive impact on patients' independence, completion of personal care tasks or engagement in therapy.

### Cognitive participation and collective action: influence of managerial authority

Managerial support for service improvements generated through EBCD/AEBCD[36] is important. In sites 1 and 2, researchers set up oversight groups including senior managers, matrons/senior nurses and staff with cross-organisational roles. The commitments of these managers meant interaction was largely through email updates or one-to-one meetings. Oversight groups could not be established at site 3 or 4 despite invitations to attend AEBCD activity. Oversight group members in sites 1 and 2 supported project activity, helped navigate complex National Health Service (NHS) Trust organisational structures and, in specific situations, provided resources to enable co-design group ideas to progress. Having identified senior managerial contacts ensured unit-based staff could activate these lines of support and translate ideas into collective action. This was more difficult in sites 3 and 4 where only the chief executive and a therapy lead engaged with AEBCD groups. Site 3 staff completed a sponsored run and the chief executive matched the sum raised. Funds (£8000) were used to redesign a day room previously only used by staff and partly to install a kitchenette for patient and family member/carer use.

In all sites, staff working outside of stroke units, including therapy service managers, matrons, patient experience managers, patient safety officers, volunteer coordinator and estates managers joined co-design groups. Although not involved in every meeting, their participation was often significant in terms of cognitive participation and collective action. NHS staff are acutely aware of resource constraints affecting their services, and most have experienced frustration at organisational barriers to improving services. Barriers identified included infection control and patient safety requirements, delays or inaction when estates work is requested and bureaucratic processes associated with including volunteers in unit-based activity. In early co-design meetings at all sites, staff that were otherwise dynamic and enthusiastic advocates for increasing

patient activity often expressed the view that such barriers were fixed and would limit what could be achieved. The perception that 'the NHS Trust would not allow' painting murals, adding shelving in patient bays, adding hot drinks facilities for patients and carers to use independently or having volunteers to support patients with social eating was pervasive. In most sites these perceptions proved to be largely inaccurate; wider hospital staff, patient and volunteer service manager typically explained how changes could be realised, and importantly, provided examples of where such changes were already operating in the same hospital.

### Coherence, cognitive participant and collective action: facilitating EBCD/AEBCD

Researchers organised staff, patient and joint events in consultation with EBCD/AEBCD-trained staff, recruited former stroke patients and carers, and ensured they could attend co-design groups. They booked accessible meeting rooms, arranged reimbursement for patients' and carers' expenses or arranged transport to and from meetings; without this level of support key EBCD/AEBCD events may not have occurred. Researchers co-facilitated staff and joint meetings; although often initiated by researchers, the joint approach built confidence in EBCD/AEBCD-trained staff. In sites 1 and 2, this increased engagement of key staff in EBCD activities. For sites 3 and 4, AEBCD-trained staff had less opportunity to work with researchers prior to joint meetings. In site 3, this appeared to have little impact on co-design meetings; whereas in site 4, team members were less confident in leading co-design meetings. At all sites, newsletters were produced by researchers or core team member to share work of co-design groups to wider staff and patient groups (online supplemental file 3).

### Collective action and reflexive monitoring: implementing and embedding co-designed changes into stroke unit practice

It took three to four co-design groups for staff to recognise potential for changes in spaces for activity and to facilitate supervised and independent activity. For most therapists and therapy assistants (TAs) increasing activity was conceptualised not only as reducing boredom and occupying patients' time but as a therapeutic opportunity. Social activity focused on lunch and breakfast groups at sites 2 and 4 provided opportunities to work on therapy goals including cognitive challenge and functional task practice. These were consistent with therapists' aims in rehabilitation, did not require working in new or different ways, and were embraced. In some sites, staff reported changes such as the 'Something About Me' board (site 2) or the 'home in the ward' personalisation of bed spaces (sites 1 and 3) made therapy more relevant as staff could draw on information made available to them through these methods. Over time, the work of implementation became more focused on embedding regularly occurring group or individually focused activity into therapists' and

TAs' work although the rate at which this occurred varied across sites.

However, post-EBCD/AEBCD observations and interviews suggested limited interdisciplinary consideration of activity promotion. Observational data indicated that initially many nurses did not develop an individual or collective understanding of EBCD/AEBCD as something they needed to participate in. This suggested that some nurse team members did not engage with the view that increasing participation in social, cognitive and physical activity was consistent with their view of what constitutes nursing work in stroke units. However, nursing staff overall expressed support for the intentions of the co-design groups but the challenges involved in nurses (in all sites) being able to attend these groups are likely to have impacted on the nurses' collective understanding and engagement in the EBCD/AEBCD process. Ward managers were aware of the challenges registered nurses (RNs) faced in taking time out of nursing care provision during 8-hour or 12-hour shifts and allocated time for RNs to attend and participate in co-design meetings. RN and HCSW attendance occurred at site 2 but not in other sites. Observations highlighted nursing staff in each site had high workloads and were frequently affected by staff shortages. Overall, there was limited evidence of RNs encouraging patient-focused social or cognitive activity. There were exceptions, two RNs and an HCSW at site 2 actively engaged in co-design work, and toward the end of AEBCD activity, nurses at site 4 routinely worked with therapists in daily breakfast groups. It is possible that as therapist and family-led activity increases in these units, greater interdisciplinary participation will follow.

## DISCUSSION

The CREATE Study process evaluation findings confirmed EBCD/AEBCD[29 31 36] was feasible to use in stroke units and that this approach facilitated development and implementation of increased activity opportunities. There were four main factors influencing engagement in EBCD/ABCD. First, the participatory approach ensured groups of staff in each unit worked directly with former patients and family/carers over a sustained period to jointly address issues key to increasing social, cognitive and physical activity levels post-stroke. Second, the structured, facilitated and time-limited nature of EBCD/AEBCD enabled each service to prioritise and agree locally appropriate changes. Third, at the outset of the study, photographic evidence gathered by researchers highlighted the typically cold, dark and clinical appearance of the stroke units. Linked to this, participants in all sites chose to work on changing physical environments before focusing on specific types of post-stroke activity. Lastly, EBCD/AEBCD made patient and family/carer experiences real in ways brief questionnaires cannot. Trigger films[29 31] confronted staff, often for the first time in their careers, with the fears, frustrations, and positive and negative reality of stroke survivors and family/carers. This proved catalytic

in securing staff commitment to bring about change. Trigger films proved powerful in raising staff awareness of ways services may need to change but these require skilled facilitation to ensure staff and stroke survivors consider the issues raised in a positive and productive way. In all units, those directly involved in EBCD/AEBCD were positive about the experience and the perceived changes underway in the units. These findings are consistent with those identified in a rapid review of outcomes of co-production studies in acute healthcare settings.[40] The shared sense of purpose and experience developed in co-design groups was consistent with what is termed a 'community of practice'[48] and led to development and implementation of innovative solutions to problems identified by patients and family/carers. Staff more peripheral to EBCD/AEBCD, particularly the larger nursing groups, reported some lack of understanding of and scepticism about the process early in the study. Organising co-design meetings and EBCD/AEBCD updates late morning or late afternoon when nursing staff were often less busy may have facilitated greater involvement and engagement. In addition, securing direct involvement of nurse ward managers in EBCD/AEBCD events was an essential element of securing high-level nursing support for change and in sustaining involvement of nursing staff in supporting change. Towards the end of EBCD/AEBCD processes, in interview and during observations most staff indicated positive views on the implementation and embedding of co-designed environmental and practice changes in sites.

Drawing on NPT's mechanisms as a sensitising device in data collection and analysis[44 45] focused attention on individual, group and organisational factors facilitating the use of EBCD/AEBCD to develop and support the implementation and potentially the longer term sustainability of changes introduced. These included engagement of EBCD/AEBCD-trained staff with the skills, expertise and resources of key staff from the wider hospital, including patient experience managers, volunteer coordinators and estates managers. Providing regular project information and updates for managers throughout the EBCD/AEBCD process was instrumental in maintaining their support. Both activities were instrumental in stroke unit staff progressing from understanding the potential of co-producing change with stroke survivors and family carers (coherence and cognitive participation) through committing personally and as staff groups to the kinds of shifts in work patterns needed to facilitate increased patient activity opportunities. These activities, together with managerial support, were linked to stimulating and supporting collective action in sites.

An additional facilitator supporting implementation of improvements in these sites was engagement with local communities. This occurred at different times and at different levels across the units; sometimes driven by family members, sometimes by patient experience managers or volunteer co-ordinators with links to colleges, charities or voluntary organisations. In each unit, there were tangible

benefits from community engagement, and community groups welcomed the opportunity to provide support to their local hospital. The early evidence suggested this engagement supported collective action and reflexive monitoring with positive feedback from inpatients and family members helping sustain staff groups' commitment to embedding improvements in routine stroke unit practice.

Our findings concur with previous EBCD/AEBCD studies in highlighting the central role of site-based facilitators in establishing and maintaining the EBCD/AEBCD process which in this study were integral to developing and implementing increased activity opportunities.[37 40 49 50] Different models are reported but in the CREATE Study researchers undertook this role at each site; they spent less time in sites than reported in other studies, and undertook facilitation alongside data collection. Given the high workload demands and clinical priorities for all stroke unit staff in the study sites, it is unlikely that the EBCD/AEBCD process would have progressed without researcher facilitation. In our view, health services using EBCD/AEBCD need to build in funded facilitator roles in each site. Drawing on NPT highlighted that implementation, in this case of both the EBCD/AEBCD process and the implementation of improvements resulting from the process was not a simple linear process and progressed at different rates in each site.

Barriers to EBCD/AEBCD implementation in all sites were broadly similar to those reported in relation to implementing complex interventions in health services.[51–53] These included staff and local organisational changes at three of the four sites. Less than optimum staffing levels combined with the high level of dependency of stroke survivors in these units made it challenging to free staff to participate in larger EBCD/AEBCD events including joint meetings and celebratory events. Similarly, although fewer staff were required to participate in co-design meetings, the frequency, duration and additional workload generated by these meetings proved challenging for EBCD/AEBCD-trained and other staff to integrate into their working day. These factors, impacted in particular, on coherence and cognitive participation; understanding the process and potential of the participatory change approach and committing to that at a personal and practice level often developed slowly in staff not directly involved in co-designing groups. In addition, EBCD or AEBCD approaches were underway for 9 and 6 months, respectively, and engaging all stroke unit staff in developing and implementing improvements proved difficult at times. Nonetheless, staff did not resist or sabotage the EBCD/AEBCD process or subsequent changes. The key factor in increasing cognitive participation and progression to collective action was the increasing visibility of changes agreed in co-design meetings. As activity opportunities were integrated into staff's daily or weekly routines, independent and family/carer or volunteer supported patient activity also increased in newly created spaces on the stroke units. At the end of the study, these changes appeared to be impacting on unit cultures and staff practice, in particular more group activities were occurring and were increasingly part of the thinking of staff, particularly therapists, but also with some nursing involvement in each unit.

This process evaluation differed from some other complex intervention evaluations in two ways. First, the use of logic models to define an intervention, its anticipated mechanisms of action and to frame research questions and methods in process evaluation are commonly advocated.[42] However, given the defined approach, stages and structured activities and mechanisms of action identified in previous EBCD/AEBCD studies, a logic model was not developed in the CREATE process evaluation. Second, the researchers conducting the process evaluation were members of the core research team rather than working independently of that team. This ensured that researchers were able to both participate in and observe EBCD/AEBCD activities. We acknowledge the limitation that in working closely with staff, former patients and family/carers, researchers were themselves part of the EBCD/AEBCD process and the design and implementation of changes they were evaluating. Recruitment of stroke survivors and family carers to participation in EBCD/AEBCD activities was good across all sites but it proved more difficult to recruit former inpatient stroke survivors to participate in post-EBCD/AEBCD evaluation interviews. So, while the process evaluation was a comprehensive study, we acknowledge that the four participating sites may not be representative of stroke units elsewhere in the UK or other countries. Lastly, the process evaluation was not designed to evaluate the longer term sustainability of interventions developed in the CREATE Study; future studies would benefit from such evaluation.

## CONCLUSION

The findings from the CREATE Study, the first of its kind in stroke services, suggest using a co-production approach was instrumental in creating conditions for locally determined former patient and family/carer-led change and innovation in service provision. It was possible to use EBCD/AEBCD in stroke units providing post-acute and rehabilitation care; this facilitated development and implementation of environmental changes and revisions to work routines which supported increased activity opportunities. NPT's mechanisms were instrumental in identifying facilitators and barriers at the individual, group and organisational levels, and attending to these will benefit future implementations of similar approaches. The introduction of EBCD/AEBCD as part of a funded research programme legitimised and supported the co-production activity in these units. However, with appropriate facilitation, managerial engagement and use of wider hospital resources, the approach could be used in other rehabilitation stroke services and similar post-acute inpatient environments.

**Author affiliations**

¹Academic Unit for Ageing and Stroke Research, Leeds Institute of Health Sciences, University of Leeds Faculty of Medicine and Health, Leeds, UK
²Faculty of Health and Social Care Sciences, Kingston University and St George's, University of London, London, UK
³Leeds Institute of Health Sciences, University of Leeds, Leeds, UK
⁴Department of Clinical Neuroscience, CCS, Monash University Melbourne and Alfred Health, Melbourne, Victoria, Australia
⁵Florence Nightingale Faculty of Nursing, Midwifery & Palliative Care, King's College London, London, UK
⁶School of Design, The Glasgow School of Art, Glasgow, UK
⁷Department of Public Health Sciences, King's College London, London, UK
⁸Faculty of Health Social Care and Education, St Georges University of London, London, UK

**Acknowledgements** The authors acknowledge the contribution of the stroke survivors, family members, stroke unit staff, volunteers, hospital managers and local community groups, and the support for the study of the four participating NHS Foundation Trust hospitals. They also wish to acknowledge colleagues who supported elements of the CREATE Study, these include: Dr Tino Kulnik who conducted pre-implementation interviews for site 1; Dr Alessia Costa who supported ethnographic fieldwork in site 1 and Carole Pound who supported co-design events in site 1.

**Contributors** The CREATE Study and embedded process evaluation were designed by FJ, DC, GR, CM, RH, GC and AM. FJ, DC, SH and KG-W collected and analysed the data. AM, CM and RH participated in some joint meetings and celebration events as part of EBCD/AEBCD cycles. DC developed the manuscript. FJ, GR, CM, AM, RH, GC, SH and KG-W reviewed and edited the manuscript. All authors read and approved the final manuscript.

**Funding** This study was funded by the National Institute for Health Research (NIHR) (under its Health Services and Delivery Research (HS&DR) Programme; grant reference number: 13-11/495).

**Disclaimer** The views expressed are those of the author(s) and not necessarily those of the NIHR or the Department of Health and Social Care.

**Competing interests** GR teaches on the EBCD training courses run by 'The Point of Care Foundation' in London. GR is one of the originators of EBCD. The remaining authors declare no other competing interests.

**Patient consent for publication** Not required.

**Ethics approval** The Health Research Authority and South East Coast Brighton and Sussex Research Ethics Committee (16/LO/0212) provided ethical approval.

**Provenance and peer review** Not commissioned; externally peer reviewed.

**Data availability statement** All data relevant to the study are included in the article or uploaded as supplemental information. No additional data are available.

**ORCID iDs**
David Clarke http://orcid.org/0000-0001-6279-1192
Ruth Harris http://orcid.org/0000-0002-4377-5063
Glenn Robert http://orcid.org/0000-0001-8781-6675

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
