## [Reviewer comments · BMJ Open]

ARTICLE DETAILS

TITLE (PROVISIONAL)	Co-designing organisational improvements and interventions to increase inpatient activity in four stroke units in England; a mixed-methods process evaluation using Normalisation Process Theory.
AUTHORS	Clarke, David; Gombert-Waldron, Karolina; Honey, Stephanie; Cloud, Geoffrey; Harris, Ruth; Macdonald, Alastair; McKeivitt, Christopher; Robert, Glenn; Jones, Fiona

VERSION 1 – REVIEW

REVIEWER	Sarina Fazio University of California, Davis, Unites States
REVIEW RETURNED	17-Aug-2020

GENERAL COMMENTS	The authors present a very detailed account of their process evaluation of the experience based co-design methods used to develop and implement activity interventions for inpatient stroke survivors as part of the larger CREATE study. While the results of this study are likely valuable to other researchers utilizing participatory-based methods for developing, implementing and sustaining inpatient interventions, there are a number of limitations of the paper that need to be addressed: 1) The paper is extremely long (6855 words) and could be considerably condensed to be more cohesive and succinct; 2) The authors mention EBCD/AEBCD and refer to “other studies/methods” utilizing similar methods but do not discuss other participatory-based research methodologies and how their results compare/contrast with those methods and or results; 3) The objective stated in the abstract, does not match with the objective presented in the paper. By describing so much of the different components of the CREATE study, the methods specific to this paper are lost and the paper is cumbersome to read; 4) The authors provide detail on the analysis process but do not provide adequate detail about who conducted the ethnography portion of the study; 5) The methods state that informed consent was obtained, does this include with patients/families/carers? Needs more detail; 6) The results need more structure with headings to guide the reader through the results specific to your process evaluation. Currently, they are very NPT specific rather than results specific. The discussion describes the main findings from the results succinctly - I would use this to restructure your results to make them more cohesive; 7) There are no study limitations presented in the main manuscript; and 8) There are a number of oversights related to abbreviations, commas, periods in the manuscript that should be revised.
--

REVIEWER	Elizabeth Lynch University of Adelaide, Australia
-----------------	--

REVIEW RETURNED	30-Aug-2020
-------------

GENERAL COMMENTS	I read the paper on the process evaluation of the CREATE co-design study in UK stroke units with interest. I found the paper clearly written and easy to follow. I think this paper provides a useful guide for clinicians and researchers about the feasibility and rationale of doing co-design in busy workplaces - the successes and the challenges. I found it really interesting how official hospital communication channels were not always available/successful to promote awareness of the EBCD project, but things like informal chats with researchers as they collected data built awareness of the project with both staff and patients/families. Also the lack of in-hospital cascade training, problems with ethics delays I think are common problems and have been highlighted clearly here. In terms of things to change in this manuscript, I have very few comments because I think it reads really well. I would encourage the authors to add a little more information in the Introduction as to why activity in stroke rehab is important, particularly since BMJ Open is a non-stroke specific journal. As written, there is an implicit assumption that lack of activity is a bad thing, yet the only reason given by the authors why inactivity is bad is that patients are bored. I would encourage the authors to present a sentence or 2 on why inactivity is thought to influence stroke outcomes. The studies presented in paragraph 2 I am not familiar with (Ellul, Newell), so it would be easier for the reader to present a tiny bit of info on where and when these studies were conducted, potentially sample sizes etc There were a small number of typos (e.g.stoke-stroke) and misplaced commas (after although from memory) which should be rectified prior to publication. For consideration - The challenges of true interdisciplinary contribution of the project came through in the Results but were not highlighted as much I though they could have been in Discussion - in particular the difficulty in involving/having contribution from nursing staff. Given the high proportion of staff made up from the nursing profession, this is a significant challenge, and one that I have faced myself in work in Australia - the need to get true buy-in and contributions/ support/ perspectives etc of nurses is really important, so if the authors had any take-home messages on what can be learnt from their experiences on how to do this better, I think that would be helpful for readers and researchers.
---

REVIEWER	Claire Stewart Australia
REVIEW RETURNED	12-Oct-2020

GENERAL COMMENTS	General Comments: Thank you for the opportunity to review this manuscript. It is an important paper describing the barriers and facilitators of using a co-design approach to change practice and a first in the area of acute stroke. My main comments relate to the difficulties in describing one embedded component (process evaluation) of an integrated larger
---

	project (CREATE). I think that the clarity of the manuscript could be improved by removing much of the CREATE methods that do not relate to this manuscript. It may also be beneficial to use a figure to describe the design of the larger project and where the methods of this manuscript fit within the larger study. Specific thoughts on each section of the paper are written below. Abstract: Lines 27-29 and 30-31. These are quantitative findings of the main study and results and do not address the aims of the process evaluation which is to understand barriers and facilitators to using EBCD/AEBCD and could be removed from abstract. Introduction: A clear justification for the study and the need for co-design approaches to develop interventions to increase the activity of stroke survivors in stroke units has been provided. There is a clear succinct explanation of the larger CREATE study findings with reference to previous study if further detail is required for the reader. The final paragraph of the introduction provides a clear rationale for the process evaluation and the gap in the wider literature. The aim of the process evaluation is clearly articulated. Methods: I found the description of the CREATE study methods and then the process evaluation methods a bit confusing. Some of the methods described are part of the larger CREATE study (e.g. behavioural mapping) and do not address the aims of the paper and are not reported in the results. I understand and commend the effort to clearly explain the embedded nature of the process evaluation by describing all the methods used in the larger CREATE study however it may be clearer to describe/illustrate the embedded process evaluation design within the larger study design (potentially using a figure). It could then be articulated that this manuscript describes only the methods related to the process evaluation and that the wider methods are reported elsewhere. The context, participants and the data collection methods only relating to the process evaluation and the aim of this manuscript could then be described. There is a clear description of process evaluation methods, NPT and the use of NPT to guide data collection and analysis. Process evaluation findings The title of this section could be changed to results. Process evaluation findings are clearly described linked to NPT with appropriate quotes as examples.
--	--

	Page 12 line 47 – May benefit from a quote to support the lack of coherence and cognitive participation among nursing staff. Page 16 line 5 - “Organising co-design meetings and EBCD/AEBCD updates at these times when nursing staff appeared less busy may have facilitated greater involvement and engagement.” This appears to be a reflection from the authors and may sit better in the discussion. Discussion Were there any key recommendations for future research/practice change using co-design methods? Any thoughts on how to overcome some of the barriers identified? Specific areas that come to mind:  • Are there any possible ways to more quickly engage staff that could not attend co-design workshops? - could shorter trigger films be shown at ward meetings as they seem to be a powerful tool in engaging staff to create change? • The significant role of the facilitator, any learnings for future studies/health services?
--	--

VERSION 1 – AUTHOR RESPONSE

Reviewer 1:	Response
The authors present a very detailed account of their process evaluation of the experience based co-design methods used to develop and implement activity interventions for inpatient stroke survivors as part of the larger CREATE study. While the results of this study are likely valuable to other researchers utilizing participatory-based methods for developing, implementing and sustaining inpatient interventions, there are a number of limitations of the paper that need to be addressed	Thank you for these general comments. We detail our responses to the comments made in the sections below.
The paper is extremely long (6855 words) and could be considerably condensed to be more cohesive and succinct	We have reduced the word count to 6235 words mainly through removal of text related to the main study methods but in other sections too (see below). In revising the manuscript we have tried to balance adding information requested by the three reviewers and Editor with the request to reduce the overall word count.

The authors mention EBCD/AEBCD and refer to “other studies/methods” utilizing similar methods but do not discuss other participatorybased research methodologies and how their results compare/contrast with those methods and or results	We acknowledge this comment. However, our aim in the study and in this paper was not to compare EBCD/AEBCD with other participatory design methods but rather to conduct a process evaluation of the use of EBCD/AEBCD in the CREATE study. To our knowledge at this time there are no published comparable studies of the use of EBCD/AEBCD in stroke services. We have identified this in the introduction (pages 3 and 4) and in the discussion (pages 15 and 16) where our study has identified more general findings (e.g. the key role of facilitators and the importance of managerial support) which are consistent with other EBCD studies.
The objective stated in the abstract, does not match with the objective presented in the paper. By describing so much of the different components of the CREATE study, the methods specific to this paper are lost and the paper is cumbersome to read	Thank you for this comment. We have ensured that the objective presented in the abstract and in the aim stated on page 4 are the same. We have removed the detailed summary of the CREATE study methods which we acknowledge may have made the reporting of the process evaluation more difficult to follow. The text which remains reports the process evaluation methods and findings specific to this paper. Following a suggestion from reviewer three we have added a new diagram (Figure 1 revised) to illustrate the embedded process evaluation design within the larger CREATE study design.
The authors provide detail on the analysis process but do not provide adequate detail about who conducted the ethnography portion of the study	We have added sentences to clarify who conducted the ethnographic elements of the study, and the duration and timing of the observations to page 7 of the manuscript.
The methods state that informed consent was obtained, does this include with patients/families/carers? Needs more detail	This information has been added to page 6 of the manuscript.

The results need more structure with headings to guide the reader through the results specific to your process evaluation. Currently, they are very NPT specific rather than results specific. The discussion describes the main findings from the results succinctly - I would use this to restructure your results to make them more cohesive	Thank you for this comment. We acknowledge that there are different ways in which process evaluations are reported, particularly those which utilise Normalisation Process Theory. In developing the manuscript we chose to use headings for the findings/results section which combined the NPT mechanisms evident and the main process evaluation findings as the related to the use of EBCD/AEBCD to increase activity opportunities in the four stroke units. The text related to each heading in the findings/results section is specific to the process evaluation. We are pleased that you consider the discussion to summarise the findings succinctly. However, we feel that reducing the focus on NPT as a means to explain our process evaluation findings would reduce the value of the paper to clinicians and researchers as highlighted by reviewers 2 and 3.
There are no study limitations presented in the main manuscript	Thank you for highlighting this. We have added text to the last paragraph of the discussion on page 17 to address this.
There are a number of oversights related to abbreviations, commas, periods in the manuscript that should be revised.	Thank you for highlighting this, we have reviewed the manuscript carefully and hopefully have addressed these issues.

Reviewer 2	Response
I read the paper on the process evaluation of the CREATE co-design study in UK stroke units with interest. I found the paper clearly written and easy to follow. I think this paper provides a useful guide for clinicians and researchers about the feasibility and rationale of doing codesign in busy workplaces - the successes and the challenges.	Thank you for these positive comments regarding the structure and content of the manuscript. It was our intention to ensure the paper was relevant to both clinicians and researchers.

I would encourage the authors to add a little more information in the Introduction as to why activity in stroke rehab is important, particularly since BMJ Open is a non-stroke specific journal. As written, there is an implicit assumption that lack of activity is a bad thing, yet the only reason given by the authors why inactivity is bad is that patients are bored. I would encourage the authors to present a sentence or 2 on why inactivity is thought to influence stroke outcomes.	Thank you for pointing this out, we had overlooked this important point. We have now added sentences to paragraph 2 on page 3 to address this omission.
The studies presented in paragraph 2 I am not familiar with (Ellul, Newell), so it would be easier for the reader to present a tiny bit of info on where and when these studies were conducted, potentially sample sizes etc.	Thank you for this comment. These were early UK studies. We have added brief additional commentary on these studies to paragraph 2 on page 3.
There were a small number of typos (e.g. stokestroke) and misplaced commas (after although from memory) which should be rectified prior to publication	Thank you for highlighting this, we have reviewed the manuscript carefully and hopefully have addressed these issues.
For consideration - The challenges of true interdisciplinary contribution of the project came through in the Results but were not highlighted as much I thought they could have been in Discussion - in particular the difficulty in involving/having contribution from nursing staff. Given the high proportion of staff made up from the nursing profession, this is a significant challenge, and one that I have faced myself in work in Australia - the need to get true buy-in and contributions/ support/ perspectives etc of nurses is really important, so if the authors had any take-home messages on what can be learnt from their experiences on how to do this better, I think that would be helpful for readers and researchers.	We agree, this is an important and challenging element of introducing change in any healthcare setting and was pertinent to the use of EBCD/AEBCD in our study. We were mindful of not increasing the word count of the paper further but to address your comment and a similar one from Reviewer three, we have added some text to paragraph 1 on page 15 of the manuscript.

Reviewer 3	Response
Thank you for the opportunity to review this manuscript. It is an important paper describing the barriers and facilitators of using a co-design approach to change practice and a first in the area of acute stroke.	Thank you for these positive comments regarding the content of the manuscript. It was our intention to ensure the paper was relevant to both clinicians and researchers seeking to utilise EBCD/AEBCD in stroke and other health services.

VERSION 2 – REVIEW

REVIEWER	Claire Stewart Australia
REVIEW RETURNED	05-Dec-2020
GENERAL COMMENTS	Clear changes made in response to reviewers comments to improve clarity of the manuscript. Accept.